# Characterization and Involvement of Exosomes Originating from Chikungunya Virus-Infected Epithelial Cells in the Transmission of Infectious Viral Elements

**DOI:** 10.3390/ijms232012117

**Published:** 2022-10-11

**Authors:** Bao Chi Thi Le, Ati Burassakarn, Panwad Tongchai, Tipaya Ekalaksananan, Sirinart Aromseree, Supranee Phanthanawiboon, Yada Polsan, Neal Alexander, Hans J. Overgaard, Chamsai Pientong

**Affiliations:** 1Department of Microbiology, University of Medicine and Pharmacy, Hue University, Hue 4200, Vietnam or; 2Department of Microbiology, Faculty of Medicine, Khon Kaen University, Khon Kaen 40002, Thailand; 3HPV & EBV and Carcinogenesis (HEC) Research Group, Faculty of Medicine, Khon Kaen University, Khon Kaen 40002, Thailand; 4Department of Anatomy, Faculty of Medicine, Khon Kaen University, Khon Kaen 40002, Thailand; 5MRC International Statistics and Epidemiology Group, London School of Hygiene and Tropical Medicine, London WC1E 7HT, UK; 6Faculty of Science and Technology, Norwegian University of Life Sciences, P.O. Box 5003 Ås, Norway

**Keywords:** chikungunya virus (CHIKV), extracellular vesicles, EVs, exosomes, cell-to-cell transmission, infectious viral elements

## Abstract

The Chikungunya virus (CHIKV) is a mosquito-borne alphavirus that affects the world’s popula-tion with chikungunya disease. Adaptation of the viral life cycle to their host cells’ environment is a key step for establishing their infection and pathogenesis. Recently, the accumulating evidence advocates a principal role of extracellular vesicles (EVs), including exosomes, in both the infection and pathogenesis of infectious diseases. However, the participation of exosomes in CHIKV infec-tion and transmission is not well clarified. Here, we demonstrated that the CHIKV RNA and pro-teins were captured in exosomes, which were released by viral-infected epithelial cells. A viral genomic element in the isolated exosomes was infectious to naïve mammalian epithelial cells. The assay of particle size distribution and transmission electron microscopy (TEM) revealed CHIKV-derived exosomes with a size range from 50 to 250 nm. Treatments with RNase A, Triton X-100, and immunoglobulin G antibodies from CHIKV-positive patient plasma indicated that in-fectious viral elements are encompassed inside the exosomes. Interestingly, our viral plaque for-mation also exhibited that infectious viral elements might be securely transmitted to neighboring cells by a secreted exosomal pathway. Taken together, our recent findings emphasize the evidence for a complementary means of CHIKV infection and suggest the role of exosome-mediated CHIKV transmission.

## 1. Introduction

The Chikungunya virus (CHIKV), a positive-sense and single-stranded RNA virus, is classified as an alphavirus in the family Togaviridae [1]. CHIKV is the pathogen causing chikungunya fever, a self-limited acute viral illness causing extreme joint pain. CHIKV infection has become a significant global health problem [2] not only related to re-emergence with numerous outbreaks but also with chronic clinical complications, such as persistent arthralgia [3,4]. Since chikungunya arthritis treatment is not specific, and prevention by vaccines is not available, it is important to understand the mechanisms of pathogenesis and development of chronic arthritis following CHIKV infection [5,6]. Many previous studies have explained the mechanism of post-CHIKV arthritis as a persistent infection with recruitment and activation of macrophages [7,8] as well as increases of many inflammatory mediators [9,10] together with the persistence of virus or viral products in the synovial tissue [11]. However, whether associated with a proinflammatory mediator, e.g., granzyme A [12], or autoimmune processes, the exact mechanism of post-CHIKV arthritis is still unclear [13,14].

Many studies have emphasized the potential role of extracellular vesicles (EVs), including exosomes involved in the pathogenic mechanisms of various pathogens. EVs are generally referred to as microparticles which can be secreted from various cells [15]. EVs are enriched in markers such as tetraspanins (CD63, CD9, CD81, and more) and other exosomal biogenesis proteins such as heat shock protein 70 and small GTPases [16]. However, the presence of protein markers was verified at different concentrations among different subtypes of EVs. In addition, the small EVs—exosomes can also interact via receptor-ligand, direct menbrance integration, or even via endocytosis internalization, which refers to a novel cell–cell transmission path way [17]. Since their discovery, several functions of EVs have been proposed, such as providing a mechanism for cell-to-cell communication [16] or involvement in tumorigenesis, antigen-specific immune responses, immune evasion, immune suppression, and immune tolerance [17,18]. Therefore, EVs can be a potential biomarker for diagnosing many diseases and play a prospective role in treatments. On other hand, choosing a proper method for the isolation of purified EVs plays a crucial role in EVs research and therapeutic analysis. Many recommended isolation methods are: differential centrifugation, density gradient ultracentrifugation, filtration, and immunoaffinity capture-based techniques [19,20]. Together with many traditional techniques, antibody-coated magnetic beads seem to be a promising and most effective specific method for the isolation and characterization of EVs [17,21,22].

Interestingly, since the budding process of EVs is similar to viral budding, RNA viruses can exploit EVs as a vehicle to transfer some viral particles or even naked viral genomes to infectious host cells [23,24]. For example, a kind of EVs called exosomes emanating from cells infected with human immunodeficiency virus type 1 (HIV-1) contained viral transactivation response elements, microRNAs, or unspliced HIV-1 species. Similarly, exosomes derived from hepatitis C virus (HCV) infected cells contained replication-competent viral RNA and even the complete HCV genome [25,26]. Although studies on exosomes derived from flavivirus-infected cells are limited, there is evidence that exosomes are involved in viral infection and transmission. Extracellular vesicles were also involved in dengue virus serotype 2 (DENV2)’s transmission from arthropod to mammalian cells mediated by interactions with an arthropod EV-enriched marker Tsp29Fb [27]. There are few studies on CHIKV intracellular replication by host endosomal sorting complex required for transport (ESCRT) [28] or by enhancement of antibody-mediated CHIKV infection which showed the infectious development and exacerbation of disease severity [29]. Additionally, a novel infectious mechanism of CHIKV has been described in which the virus is concealed in apoptotic blebs assisting in the invasion and infection of neighboring host cells as well as evading host responses [30].

We have limited documentation of any reports on the potential contribution of EVs to the pathogenesis of CHIKV. Therefore, our study aimed to characterize extracellular microvesicles such as exosomes secreted from CHIKV-infected epithelial cells and determine their role in CHIKV infection and transmission.

## 2. Results

### 2.1. Susceptibility Profiles of the Established Epithelial Cell Lines to CHIKV Infection

Although the muscle satellite cells, fibroblasts, endothelial cells, and macrophages have been permissive to CHIKV infection, several studies have demonstrated the susceptibility of various epithelial cells to CHIKV infection [31,32,33,34]. To this end, we firstly evaluated the sensibility of Vero cell lines, which are continuous tissue culture cell lines derived from African green monkey kidney epithelial cells for the infection of the clinical CHIKV ROSS C6D4 strain. Moreover, the kinetics of CHIKV growth was perceived in HeLa cell lines, which is an HPV18-positive human cervical cancer cell lines and generic permissive arthropod cell lines, C6/36. As demonstrated in Figure 1, CHIKV at the multiplicity of infection (MOI) 5 facilely infected the Vero cell line with significantly increased transmembrane glycoproteins E1 mRNA loads at 72 h and 96 h post-infection (h.p.i.) in comparison with 48 h.p.i. (Figure 1A). The highest CHIKV loads have arisen between 12 h.p.i. to 24 h.p.i. in C6/36 (Figure 1B), while HeLa cells were observed at 48 h.p.i. (Figure 1C). Taken together, the three cell lines are progressively susceptible to CHIKV ROSS C6D4 strains and exhibit strain-specific differences in a time-dependent manner and an enormity of infectious viral load levels.

### 2.2. CHIKV Specifically Facilitates the EV Population

To investigate whether epithelial cells secrete EVs, including exosomes, and if mosquito-borne alphaviruses use those exosomes as one of the pathways of their transmission, we next characterized the exosomes that were isolated from CHIKV-infected epithelial cells and analyzed the existence of viral genomic material and their related proteins. Interestingly, Vero cell-derived exosomes carried CHIKV transmembrane glycoproteins E1 mRNA loads in a manner of time, with progressively increased viral mRNA levels at 72 h.p.i. when infected with 5 MOI of CHIKV, compared with the early experimented time points of 24 and 48 h.p.i. (Figure 2A). Depending on the increased viral loads, we, therefore, selected the time point at 72 h.p.i. for the isolation of exosomes from CHIKV-infected Vero cell lines.

Much of the exosomes were purified using a commercially available exosome isolation reagent and we further sequestered the small vesicle subset with an anti-CD63 Dynabeads™ method. Detailed protocols for exosome isolation in this study are shown as schematics in Appendix A. Using a transmission electron microscopy (TEM) technique, we performed the negative staining on Vero cell-derived CD63^+^ exosomes originating from both the CHIKV infection and the parental cells revealing the occurrence of those EVs subset with the distributed size from 30–250 nm in diameter (Figure 2B), which were similar to exosomes that were quarantined from other mammalian epithelial cells, including human epithelial cells [35,36,37]. Calculating the total number of isolated exosomes from uninfected cells exhibited no significant differences in comparison with the 5 MOI CHIKV infection at 72 h.p.i. (Figure 2C). In addition, we quantifiably analyzed the population of the isolated heterogeneous Vero-derived exosomes and indicated that the size range of 50 and 100 nm (in diameter) was found to be the exalted population in both the uninfected and the CHIKV-infected groups (Figure 2D,E). Nevertheless, the small EVs originating from CHIKV-infected Vero cells with the 50–100 nm and 100–150 nm sizes had progressively increased and tended to decrease population percentage in the uninfected control (Figure 2D,E). Consequently, a lower population of the large Evs, in the size range of 200–350 nm, was also observed in CHIKV infection in comparison with the uninfected group, implying that the production of smaller-sized Evs might be stimulated by CHIKV (Figure 2D,E).

Since the physical transmission electron microscope (TEM) and Dynamic light scattering (DLS) characterization indicated the high pure quality of the exosomes, we, therefore, estimated the total amount of proteins of those Evs by Bradford protein assay (BCA). The purified small Evs had the total amount of proteins in a manner of cell numbers plating, with linear increase concentration at the indicated time point (Figure 2F,G). Remarkably, the concentrations of total protein from the CHIKV-infected cells were not significantly different in comparison with the uninfected controls (Figure 2F,G), correlating with the data on the total number of small EVs between the CHIKV-infected and the uninfected groups (Figure 2C).

To further exemplify these EV subgroups, we then determined the expression of CD63 and CD9 which are the well characterized EV biogenesis markers. Our immunoblotting revealed both CD63 and CD9 signals were detailed in exosomes that were isolated from Vero cells with or without CHIKV infection (Figure 2H). We also corroborated that cellular cytochrome-C was not captured in these exosomes (Figure 2H), reminding us of no contamination from other types of EVs.

Collectively, these data suggested that CHIKV specifically mediated the excretion and cargo of the EVs’ population consistent with the features of those EVs’ subsets.

### 2.3. CHIKV RNA Elements Are Encapsulated by Secreted Exosomes and Are Infectious

Because we demonstrated that discrete exosomes might be mediated by CHIKV infection, we subsequently evaluated whether these vesicles contain the infectious viral genomic element. To this end, a reverse transcription PCR (RT-PCR) method described by Stapleford et al. in 2016 [38] was utilized to verify the full length of the CHIKV genome. As we expected, the RT-PCR detection documented the overlapped oligonucleotide amplicons that consisted of the entire CHIKV genome sequence in Vero cell-derived exosomes (Figure 3A). Using the quantitative real-time PCR (qRT-PCR), we also illustrated both CHIKV E1 mRNA in exosomes derived from MOI 5 infected Vero cells at 72 h.p.i. (Figure 3B). We subsequently determined whether these exosome-contained viral genomic elements are infectious and if CHIKV might be transmitted to neighboring cells via exosomes. We detected the cytopathic effects (CPEs) and the progressively increasing trend of CHIKV loads in a time-dependent manner in the naïve Vero cells that were exposed to the exosomes from the CHIKV-infected Vero cell lines (Figure 3C,D). In addition, in the calculation of the plaque-forming unit (PFU), we were able to demonstrate the high CHIKV titer (Figure 3E) and CHIKV particles in the cell culture medium (Figure 3F) in the exosomes-exposed naïve Vero cells but not observed in the exposure of exosomes from the CHIKV-uninfected group.

As demonstrated, small EVs containing genomic material of CHIKV are infectious. How do they internalize a cell? We might not rule out the possibility of some viral particles being coexistent either outside or inside of those EVs. Because no CHIKV virions were detected either inside or outside of these isolated EV parts (Figure 2B), therefore, we suppose this to be inconsequential, suggesting that viral RNA was feasibly appropriate to temper the infection of naïve recipient cells. Altogether, we certified that an infection of CHIKV might be transmitted by those small EVs which could be distinct from true virions.

### 2.4. CHIKV RNA Elements and Proteins Are Securely Contained Inside the Exosomes

Since viral RNA could be extracellularly originated in a non-exosome-encapsulated form, so we aimed to address whether this viral genomic material was encompassed within Vero cell-derived small EVs. To assess this assumption, we examined if CHIKV genomic RNA was attaching to the exosome outside and could then be conveyed to the naïve recipient cells. We did not obtain significant differences in CHIKV loads of naïve Vero cells that were incubated with exosome formulated from 5 MOI CHIKV infection at 72 h.p.i. and/or RNase A treatment. Naïve Vero cells incubated with exosomes derived from CHIKV-uninfected cells and RNase A were used as the model of an internal-based line control (Figure 4A). As the deep inside of the viral genome and the inability of RNase A to be accessible in the CHIKV envelope, only RNase A treatments might not be strong enough to verify our hypothesis. Thus, we include the Triton X-100 reagent in the RNase A treatment assay. The incubation of Triton-X before RNase A treatment on these exosomes also yielded an undetectable CHIKV RNA in the naïve Vero cells (Figure 4B). In addition, we performed another RNase A-treatment assay, independently using MOI 5 of laboratory CHIKV stocks accompanied by infectious exosomes for comparison. As would be anticipated of the encapsulated CHIKV genomes, we achieved imitated outcomes with no differences in viral loads in naïve Vero cells after incubation with exosomes or CHIKV stocks prepared from RNase A-treated or untreated groups (Figure 4C).

To test if the viral proteins were also carried inside the Vero-derived EVs, that were not found in the PBS suspensions, we further performed and fabricated neutralizing antibodies against the CHIKV assay in order to confirm that some viral proteins were not organizing outside of the exosome as a contaminant. As demonstrated in Figure 4D, no significant differences were perceived in CHIKV loads between the treatment of exosomes derived from 5 MOI CHIKV-infected Vero at 72 h.p.i. with either anti-CHIKV IgG from patient plasma or with relevant isotype control and the untreated controls (Figure 4D). We also verified the MOI 5 of laboratory CHIKV strains as a control and exhibited a reduction of CHIKV loads in Vero cells in comparison with the isotype antibody-treated or untreated controls (Figure 4E). This information supported a positive effect for the small group and suggested that some CHIKV proteins were securely contained inside the CHIKV-derived exosomes.

To corroborate that those viral antigenic proteins were contained inside the small EVs, we performed ELISA with anti-CHIKV IgG, where higher loads of CHIKV were detected when exosomes were permeabilized with Triton X-100 in comparison with the untreated group. The inside viral protein was released upon lysis of the exosome lipid bilayer, thereby occasioning in augmented detection of CHIKV protein with the Triton X-100 permeabilized treatment (Figure 4F). Overall, our experimental data indicate that CHIKV RNA and proteins were securely contained inside the epithelial-derived exosomes.

### 2.5. CHIKV Infectious Elements Are Transmitted to Neighboring Cells through Exosomes

To further understand the influential role of infectious exosomes in the infection, replication, and transmission of CHIKV, we, therefore, performed the inhibiting assay of exosome biogenesis. Dihydrochloride hydrate substance, GW4869, is a selective cell-permeable neutral sphingomyelinase inhibitor, affecting the biogenesis and releasing of exosomes from the cells used in this experiment. Treatment of Vero cells with GW4869 at various concentrations (10, 15, and 20 μM) followed by CHIKV infection at MOI 5, for 72 h.p.i. significantly decreased CHIKV loads in the small EV fraction in comparison with the DMSO control (Figure 5A). At 72 h.p.i. of incubation of naïve Vero cells with exosomes, 5 MOI CHIKV-infected Vero cells treated with 15 μM of GW4869 disclose not only a decrease in CHIKV viral replication but also CHIKV transmission to naïve Vero cell in comparison with the DMSO treatment (Figure 5B). We consequently performed a transwell assay on exosomes- (derived from 5 MOI CHIKV-infected Vero cells at 72 h.p.i) infected Vero cells that were incubated with or without treatment of 20 μM of GW4869 inhibitor (the upper chamber) and naïve Vero cells (the lower chamber). Similar results were obtained and showed that the newly produced exosomes were able to transmigrate and infect naïve Vero cells in the lower chamber (Figure 5C). Taken together, these data implied that the direct participation of exosomes with genomic elements and proteins of CHIKV was not only involved in CHIKV infection in the mammalian host cells but was also necessary for the transmission of infectious viral elements to the neighboring cells.

## 3. Discussion

Currently, studies of CHIKV’s pathogenic mechanism mainly focus on the interaction between the host and viral particles. Among those theories, the role of extracellular vesicles has only been investigated in host–pathogen relationships [26,39]. As far as we know, the present study is the first to characterize EVs derived from CHIKV-infected Vero cells.

Firstly, we found that the CHIKV infection stimulated the release of EVs which ranged from 50 nm to 250 nm Under TEM, the vesicles collected after purification by Invitrogen™ Dynabeads™ Exosome-Human CD63 isolation kit were completely clear and easy to identify, compared with indiscriminate objects from the pellets without the purification steps. We also confirmed the presence of these purified small EVs by CD63 exosomal marker and no cellular contamination by cytochrome C marker. These results showed the effective purification of the CD63 antibody-coated Dynabeads in helping to separate the viral particles from EVs. There are few studies on the infectious mechanisms by which CHIKV utilizes apoptotic blebs and infects macrophages [30]. During CHIKV replication, various host ESCRT factors accumulated via hepatocyte growth factor-regulated tyrosine kinase substrate (HGS), involving replicating CHIKV genomic RNA and post-translation processes [28]. However, no data on the connection between CHIKV and EVs have been reported. By observing heterogeneous vesicles, we investigated the presence of some kinds of subtypes of EVs (50 nm to 250 nm) in the pellets isolated from CHIKV-infected cells. The isolation of different subtypes of EVs was also reported from some recent flaviviruses in which the EVs could be classified as large (90–120 nm) or small (60–80 nm) [40]. In addition, a study on the potential ability of the Zika virus to use the EV tracking pathway showed increased production of small EVs from a Zika-infected Vero cell line [41].

Next, we tried to find properties of those small EVs which could contribute to the pathogenesis of CHIKV. We found genomic CHIKV RNA in the purified exosomes RNAse A-treated but not in the exosomes from the CHIKV-uninfected samples. The detection of CHIKV genes from the exosome pellets isolated from the CHIKV-infected Vero cells indicated that the exosomes could contain viral RNA. This phenomenon was easily explained by the budding biogenesis pattern of EVs, as it occurs when the virus is released from an infected cell [16,42]. Besides, the large size of the isolated vesicles (from 50 nm to 250 nm) could be a good vehicle for the internalized CHIKV and used as a mechanical way for transmission to other cells. These data suggested that exosomes released from infected cells may contain some parts of CHIKV’s full-length genome. Moreover, the time point that we optimized for exosome isolation at 72 h.p.i. was supposed to be the CHIKV life cycle. The stage, ranging from 8–18 h post-entry, was able to be observed in vitro when the CHIKV’s infected cells started to show cytopathic effects and found viral particles in many parts of the host cells. This optimal time also showed mature viral virion budding from the host cell’s membrane [43,44]. Therefore, the exosomes securely contained the genetic parts of CHIKV, helping replicate future CHIKV-exosomes in target cells. Our findings shared the same ideas and result in terms of localizing viral nucleic acid in exosomes from cells infected by several viruses, including hepatitis C [45], hepatitis B [46], respiratory viruses species [47,48], herpes simplex virus 1 [39], and even many viruses in the Flaviviridae family such as dengue virus [27] or Zika virus [49].

Another finding was that exosomes from CHIKV-infected cells showed neutralizing activity in CHIKV-positive human IgG plasma. Our ELISA assay results showed the same neutralizing pattern and the concentration-dependent effect between isolated exosomes and the free CHIKV ROSS strain. The above information emphasized the impact of CHIKV-positive IgG plasma, which might influence the immunogenic parts of CHIKV internalized in exosomes during the virus’ replication cycle in the host cell. Previous studies have shown that parts of CHIKV could be involved in the immunogenic and neutralizing processes. Thus, the modification of the nsP2 protein could reduce CHIKV replication [50], while the depletion of the capsid protein did not affect the assembly of infectious CHIKV particles [51]. Therefore, the role of envelope proteins E1 and E2 was not only involved in fusion and binding [52] but was also influenced by the host’s immune response [52,53,54,55,56,57]. Our above results demonstrated the presence of both nonstructural and envelope proteins, especially the E1 protein inside the exosome. Therefore, we once again can conclude that the isolated exosomes contain at least the E1 part of envelope CHIKV proteins that the anti-CHIKV human IgG antibodies could mediate. Since the anti-CHIKV human IgG antibodies’ titer was not high enough, we saw incomplete neutralizing activity in both exosomes and the CHIKV ROSS strain. Even though EVs could be affected by the human immune response, the internalized envelope protein might enhance the binding and fusion of exosomes to adjacent cells, thereby facilitating the spread of infection. The same phenomena could also be seen in the Newcastle disease virus-producing exosomes containing the viral NP protein promoting its infection [58] or shedding EVs from herpes simplex virus 1, which shields the virus from neutralizing antibodies [39]. Our follow-up experiments will assess whether the E2 protein is presented in the EVs since this protein is immunodominant [52] and interacts with human-neutralizing antibodies [56,57]. We assumed that the exosomes were indeed derived from CHIKV-infected Vero cells despite the above limitation. Besides, those exosomes internalized some parts of CHIKV or maybe the whole virus during endocytosis.

Since transmission through small EVs has been established in the hepatitis virus [45,46], the herpes simplex viruses [39], and the dengue virus [27], the following experiment confirmed the isolated small EVs’ infectivity in susceptible neighboring cells. Our results indicated the transfer of CHIKV to uninfected Vero cells through the presence of a cytopathic effect as well as the ability to establish productive infections. In addition, the high titer of the virus released from the infection assay of the exosome, suggests its infectious capacity could equal that of the free CHIKV ROSS strain. Furthermore, our embedded TEM found intact CHIKV virions in the supernatant and the cell lysate. This information confirmed that the small EVs brought infectious viral particles that produced new mature virions in other cells.

A question arising during our study was the relative importance of small EVs in the acute and chronic phases of the infection. It is known that the acute phase of CHIKV infection is defined as not less than 3 months after the onset of the disease [6]. This phase is consistent with strong host antiviral type I interferon (IFN) response, IgM antibodies, and neutralizing IgG antibodies in the second week of infection [6,59]. In addition, the acute phase is also represented by the recruitment of NK cells, macrophages, and monocytes at the site of infection [59,60]. Under the various immune mechanisms of viral clearance, small EVs can survive by internalizing inside those immune cells. After that, these EV-infected immune cells then migrate to other parts of the host body or even to the privileged site such as the synovial tissue. Therefore, EVs can play a role in spreading CHIKV and the persistence of the virus in synovial tissues that can induce the localized inflammation of chronically infected patients. On the other hand, in the case of a weak immune response, our PRNT assay showed an incomplete neutralizing activity of EVs (data not shown). We have some explanations for this observation. The first is related to the low titer of neutralizing antibodies obtained from our human plasma samples. The second is the ability of small EVs to escape from the immune response by distinct mechanisms such as modifying the binding epitopes of the antibodies on CHIKV envelope glycoproteins. In addition, exosomes can act as enhancement factors that lead to antibody enhancement, promoting entry into susceptible host cells and increasing virus replication. However, further experiments are needed to confirm such theories.

## 4. Materials and Methods

### 4.1. Cell Lines and Cultures

Three kinds of established epithelial cell lines, including mammalian and insect cells, were used in this study. Vero cells [61]—a common continuous tissue culture cell line derived from African green monkey kidney epithelial cells (a kind gift from Prof. Pilaipan Puthavathana, Faculty of Medical Technology, Mahidol University, Bangkok, Thailand) and the cervical cancer-derived human papillomavirus- (HPV18) positive cell lines, HeLa [62], were grown in high glucose Dulbecco’s modified Eagle’s medium (DMEM) (GIBCO™; Life Technology, Carlsbad, CA, USA). *Aedes albopictus* (C6/36) cells [63] were cultivated in Leibovitz’s L-15 Medium (GIBCO™; Life Technology, Carlsbad, CA, USA). All cell lines were supplemented with 10% inactivated fetal bovine serum (FBS) (GIBCO™; Life Technology, Carlsbad, CA, USA) together with a standard prepared mixture concentration of antibiotic gentamicin (40 µg/mL), streptomycin (100 µg/mL), penicillin (100 U/mL), and fluconazole (5 ng/mL) in a humidified atmosphere with 5% CO_2_ at 37 °C.

### 4.2. Preparation of Extracellular Vesicle-Depleted FBS

EV-depleted FBS was used in all experiments on the isolation of EVs, including exosomes. FBS was centrifuged at 100,000× *g* for 16 h in a 70Ti fixed-angle titanium rotor (Beckman, Fullerton, CA, USA) and then filtered through a 0.22-µm filter.

### 4.3. Propagation of Laboratory CHIKV Stock

All experiments were performed using the CHIKV ROSS C6D4 strain [64]. Viral stocks were performed in Vero cells and plaque assays were used to determine the titers of the harvested virus. To produce, harvest, and enumerate CHIKV stocks, we followed the protocol previously described by Chalaem et al. in 2016 [65].

### 4.4. Infection of CHIKV to Vero Cell Line

Vero cells at a density of 1 × 10^5^ cells were grown in each well of 12-well plates for 24 h before CHIKV infection. The cells were infected with CHIKV at 5 of the Multiplicity of Infection (MOI 5). The infected cells were collected at various time points, including 24, 48, 72, and 96 h post-infection (h.p.i.) and processed for RNA extractions. There were increased CHIKV loads at 72 h.p.i., in Vero cells, we, therefore, considered this time point for all downstream experiments.

### 4.5. Exosomes Isolation and Enrichment

To prevent the contamination of the isolated EVs with other extracellular vesicles originating from FBS, EV-depleted FBS was used in this experiment. A total of 30 mL of conditioned culture medium was collected and centrifuged at 300× *g* at 4 °C for 10 min to eliminate cell pellets. The collected supernatant was centrifuged at 2000× *g* at 4 °C for 30 min to eliminate cell debris and dead cells. The EVs, including exosomes, were isolated by using a Total Exosome Isolation Kit (Thermo Fisher Scientific, Waltham, MA, USA), according to the manufacturer’s protocols. Briefly, the prepared medium was homogeneously mixed with the 0.5 volume of total exosome isolation reagent and was incubated at 4 °C overnight. The mixture was then centrifuged at 10,000× *g* at 4 °C for 60 min. The pellet was resuspended with 1X PBS buffer. For further experiments, the enriched exosomes were divided into small aliquots and frozen at −80 °C.

### 4.6. Exosome Purification

For further exosome purification, Invitrogen™ Dynabeads™ Exosome-Human CD63 isolation kit (Invitrogen, Waltham, MA, USA) was used to separate small EV subpopulations following the manufacturer’s instructions. In brief, the Dynabeads magnetic beads, coated with the primary monoclonal antibody specific for the CD63 membrane antigen, were incubated with samples overnight, and captured exosomes were magnetically separated for downstream applications. The purified exosomes were now ready for physical characterization, RNA, and protein extraction as well as the downstream infection experiments.

### 4.7. Physical Characterization of Isolated Exosomes

To determine the size and number of the purified exosomes, we explored the distribution of exosome particles with Dynamic light scattering (DLS) instruments (Zetasizer Nano ZS; Malvern Panalytical, Worcestershire, UK). The data outputs were then analyzed using Zetasizer software version 7.13 (https://www.malvernpanalytical.com/; accessed on 22 August 2018).

For the visualization of small EVs and viral particles, the transmission electron microscope (TEM) was performed. As modified by Hurwitz et al. in 2017 [66] described, the carbon-coated copper electron microscope grids (Ted Pella, Redding, CA, USA) were adsorbed with a 10 µL of exosomes solution that contained 1X PBS for 2 min at room temperature (RT). Following the incubation, the excess solution was carefully removed using Whatman paper by touching only the grid’s outer ring before being stained with 10 µL of 4% uranyl acetate for 1 min. The stained grid was air-dried at room temperature until completely dry before the examination at an acceleration voltage of 80 kV under TEM (TECNAi G2; FEI Company, Hillsboro, OR, USA). The CHIKV virion was observed in the samples for five times on five different grids.

For CHIKV virion visualization, the supernatant and cell pellets collected from the EV-infected assay were fixed with 2% glutaraldehyde and 2% paraformaldehyde on ice and stored overnight in a refrigerator. The samples were washed four times with buffered solution, for 15 min each time, before being subjected to postfix with osmium tetraoxide for 2 h. Samples were washed an additional two times with buffer solution for 15 min and then each and all osmium tetraoxide debris was removed. The samples then went through four dehydration steps using ethyl alcohol concentrations of 30%, 50%, 70%, and 95%. Finally, the samples were covered with propylene for 10 min before embedding with propylene overnight and observed under the TEM as described above.

The visualization procedure under TEM was developed and performed at Electron Microscopy Unit, Faculty of Medicine, Khon Kaen University.

### 4.8. Determination of Exosomes Markers by Immunoblotting

To avoid any interfering reagents that could affect the captured exosome pellets, the freeze–thaw method was used for total exosome protein extraction. The extraction process consisted of two or three repetitions, quickly freezing the pellet with liquid nitrogen for 5 min and thawing combined with 5 min vortexing. After 10 min of centrifugation at 10,000× *g*, the obtained clear supernatant was used for protein measurement and exosomal marker detection by immunoblotting technique.

As described by Burassakarn et al. in 2021 [67], CD63 and CD9 proteins were used as exosomal markers and cytochrome C was used as a marker for cellular contamination control. The primary antibody used included purified mouse anti-human CD63 (ab59479, Abcam, Cambridge, UK) or rabbit anti-CD9 (D8O1A, Cell Signaling Technology; Danvers, MA, USA) was diluted at a 1:1000 ratio, and the purified mouse anti-Cytochrome C (6H2.B4; BD Biosciences, Berkshire, United Kingdom; diluted at a 1:250 ratio) under reducing conditions as recommended by the manufacturer was used in this study. An equal number of exosomes were mixed with loading dye, incubated at 95 °C for 5 min, and cooled down at room temperature for 5–10 min before loading on an SDS/PAGE gel.

Polyacrylamide gel electrophoresis was performed. Proteins on the gel were transferred to a nitrocellulose blotting membrane (Amersham Protran, Little Chalfont, UK) and the protein-transferred membrane was blocked by shaking in 5% skim milk in 0.1% Tween 20/1X PBS for 60 min at RT. A primary antibody diluted in 5% skim milk/1X PBS was added to the membrane and incubated at 4°C overnight. Then, the membrane was washed three times with a washing solution containing 0.1% tween 20 in 1X PBS for 15 min at RT. Goat anti-mouse IgG antibody, horse-radish peroxidase (HRP) conjugate (Merck KGaA, Darmstadt, Germany), diluted 1:1000 in 5% skim milk/1X PBS was used as a secondary antibody. The membrane was incubated with the secondary antibody for 1 h at RT and washed three times with a washing solution for 10 min at room temperature. According to the manufacturer’s instructions, signal detection was performed using Enhanced Chemiluminescence (ECL) prime Western Blotting Detection Reagent (Amersham, Little Chalfont, UK). Signal detection was imaged by using the image Quant™ LAS 600 instrument (GE Healthcare, Chicago, IL, USA). The membrane was exposed for 5–10 min.

### 4.9. Detection of CHIKV Genomic Materials in the Cells and Exosomes

To remove all the excess contaminated nucleic acid during purification steps, an additional RNase A treatment step was performed using 5 μg/mL RNAse A (stock 4 mg/mL) (Omega, Life Science, CO, USA) at 37 °C for 1 h.

According to the manufacturer’s guidelines, the total RNA of exosomes from either CHIKV-uninfected cells or exosome-derived CHIKV-infected cells were extracted using TRIzol^TM^ LS reagent (Invitrogen, Waltham, MA, USA) protocol. The RNA pellet was resuspended in 20 µL of RNAse-free water. Then the quality of the isolated RNA was analyzed using the NanoDrop™ 2000/2000c spectrophotometer (ThermoFisher Scientific, Waltham, MA, USA). Thereafter, total RNA was reverse-transcribed into cDNA using 1 µL random hexamer primers and SuperScript™ III First-Strand Synthesis System (Invitrogen, Waltham, MA, USA), following product instructions. The cDNA was stored at −20 °C until further analysis. Following cDNA synthesis, the entire CHIKV genome was amplified by specific primers set (Appendix A) using the standard reverse-transcription-polymerase chain reaction (RT-PCR) assay [38]. The PCR reaction was carried out using 0.625 U/µL of Phusion™ High-Fidelity DNA Polymerase (ThermoFisher Scientific, Waltham, MA, USA), 0.1 mM dNTPs, 1X 10X PCR buffer, 1.25 mM MgCl_2_ and 10 µM of each forward and reverse primer, and 4 µL of cDNA in total 25 µL reactions. The RT-PCR reaction was performed using a Bio-Rad C1000 thermal cycler (Bio-Rad, Hercules, CA, USA). Five microliters of PCR product were subjected to 1% agarose gel electrophoresis, stained with ethidium bromide, and visualized under an ultraviolet Gel Doc transilluminator (Bio-Rad, Hercules, CA, USA).

### 4.10. Triton X-100 Treatment of Small EVs

Exosomes were treated with RNase A (5 μg/mL) either with or without Triton X-100 (0.1%) for 15 min. After adding RNase inhibitor, total RNAs were isolated by TRIzol^TM^ LS reagent (Invitrogen, Waltham, MA, USA) protocol, and CHIKV RNA was quantified by qRT-PCR as described below.

### 4.11. Exosomes Infectious Assay

Vero cells were seeded on a 24-well plate at a density of 1.5 × 10^5^ cells/well. The prepared Vero cells were inoculated with 200 µL of each exosome from either the no-infection or CHIKV-infected cells. The plate was incubated inside the biosafety cabinet at the indicated time allowing exosome absorption. Then, the excess medium was removed and replaced with a fresh maintenance medium with 2% exosome-depleted FBS before incubation for 2 h at 37 °C in 5% CO_2_. The plate was observed for the presence of cytopathic effect (CPE) and supernatant and cell lysates were collected for virion detection by TEM and determination of viral infectious titer using standard plaque-forming unit assay (PFU).

### 4.12. Analysis of EV Transfer CHIKV RNA Elements and Proteins

To determine the CHIKV mRNA loads in both cells and exosomes, the qRT-PCR was performed. We used the designed specific primers to detect the *CHIKV E1 gene* (Appendix A). The β-actin primers were used in the qRT-PCR normalization. The rate of CHIKV infection in each tested group was indicated by the ratio index of the *CHIKV E1 gene* divided by the *β-actin gene*. The normalization of total RNA loads was performed in the case of some experiments with undetectable amounts of β-actin in exosome experiments. qRT-PCR was performed using the SsoAdvanced™ Universal SYBR^®^ Green Supermix (Bio-Rad, Hercules, CA, USA). A 10-fold serial dilution starting from the known quantities’ plasmid of the *CHIKV E1 gene* or the *β-actin gene* fragments was used in the establishment of a standard curve.

A PFU assay on Vero cells was performed to determine the infectious titer and capacity of CHIKV replication as well as the formation of infectious plaques in suspected CHIKV-derived small EVs. The PFU assay was based on a previous protocol with some modifications [68]. Briefly, Vero cells were seeded at 2 × 10^5^ cells/well, then cells were grown in 24-well tissue culture plates until reaching 80–90% confluent monolayer within 24 h. The culture medium was removed and the cells were inoculated with 200 µL of exosome containing unknown PFU of CHIKV viral genome culture supernatant, prepared at 10-fold serial dilutions in DMEM medium. After incubating cells at 37 °C for 2 h allowing for virus adsorption by plate and virus mixing every 30 min, cells in each well were gently overlaid with 500 µL pre-warmed 1.5% carboxymethyl cellulose (CMC) (Sigma, Chicago, MO, USA) mixed with DMEM. The plates were incubated at 37 °C with 5% CO_2_ and checked daily for CPE or plaque formation. After 48 h, the plate was fixed and stained with a mixture of crystal violet staining solution in ethanol and formaldehyde [69]. Plaques were counted and calculated to determine CHIKV viral titer in PFU/mL. All experiments were undertaken independently in a duplicate assay of titer.

To determine the neutralizing activity of antibodies against CHIKV and a small EV fraction from the CHIKV-infected Vero cells, we used plasma samples that presented a high titer of CHIKV-positive human immunoglobulin G (IgG) plasma and IgG-negative control samples from patients as an isotype control. CHIKV ROSS strain C6D4 was also used as the positive control. The titer of small EVs collected from supernatants of CHIKV-infected cells and the CHIKV ROSS strain was adjusted with MOI 5. The human plasma was mixed and incubated with an equal volume of small EVs or CHIKV ROSS strain at 37 °C, 5% CO_2_ for 1 h. Isotype controls using confirmed IgG human serum samples were also included. Thereafter, the mixture was layered on a 24-well plate containing 80–90% confluent monolayer Vero cell 2 × 10^5^ cell/well. After allowing the virus’ adsorption for 2 h, the inoculum was removed and replaced with a new complete medium. The plate was incubated at 37 °C, 5% CO_2_ for 72 h. Until the CPEs were observed, Vero cells were harvested and further processed for the detection of CHIK loads.

To determine the CHIKV proteins in the isolated exosomes, ELISA was performed. The equal amount of total proteins (estimated by BCA method) either from 0.1% Triton-X-100-treated or untreated samples derived CHIKV-infected or -uninfected Vero cells were coated in MaxiSorp™ flat-bottom 96-well plates (Nunc™, Thermo Fisher Scientific, Waltham, MA, USA) for overnight at 4 °C. Bovine serum albumins (BSA, Sigma-Aldrich, St. Louis, MO, USA) were used in the sample blocking step before being incubated with a high titer of CHIKV-positive human immunoglobulin G (IgG) plasma from our previous study [70] for 1 h. The secondary antibody, an HRP-conjugated mouse monoclonal anti-human IgG (HP-6017, dilution = 1:10,000; Merck KGaA, Darmstadt, Germany) was applied to the reaction and incubated for 1 h. According to the manufacturer’s instructions, SureBlue™ TMB 1-Component Microwell Peroxidase Substrate (SeraCare, Milford, MA, USA) was added to the well followed by the TMB Stop Solution (KPL; SeraCare, Milford, MA, USA). Using an absorbance microplate reader (Tecan™, Männedorf, Switzerland), optical density at 450 nm was measured and then analyzed with Magellan™ software (Tecan™, Männedorf, Switzerland) from triplicated samples in three independent experiments.

### 4.13. Inhibition of Exosome Production by GW4869 Inhibitor

For studies of exosome inhibition, we used a commercial selective cell-permeable GW4869 inhibitor for the Neutral Sphingomyelinase (N-SMase) (Sigma-Aldrich, St. Louis, MO, USA). Vero at the density of 2 × 10^5^ cells were seeded in a complete DMEM medium for 24 h. The day after, the GW4869 inhibitor with indicated doses of 5, 10, 15, or 20 μM was treated with the Vero for 4 h, followed by 5 MOI CHIKV infection for 72 h and 24 h in case of viral mRNA loads and immunoblotting analysis, respectively. A similar volume of DMSO treatment was used in the control groups. For analysis of the viral kinetics infection of infectious exosomes, Vero cells were treated with CHIKV-derived exosomes isolated from Vero cells treated with 15 μM of such inhibitor. DMSO was used as vehicle control. At 48 h.p.i., viral loads were determined from the cell lines by the qRT-PCR method.

### 4.14. Transwell Assay

To analyze the transmigration of infectious small EVs released from infected Vero cells to naïve Vero cells, transwell assays were performed. Sterile, 12 mm insert size polycarbonate transwell^®^ inserts with 0.4 μm microporous membrane pore size (Corning Inc., Corning, NY, USA) were used. Naïve Vero cells were seeded in 12-well plates (lower chamber) at the density of 2 × 10^5^ cells accompanied by the plated Vero cells at the density of 2 × 10^5^ in the inserts (upper chamber). At 24 h post-cultivation, Vero cells were either infected with CHIKV-derived exosomes or with laboratory CHIIKV stock prepared from cultured Vero cells supernatants (14 days post-infection). Following the 24 h post-infection either with exosomes or CHIKV stock, the inserts were transferred to naïve Vero cells in 12-well plates. Exosomes containing CHIKV RNA and proteins originating from Vero cells were authorized to migrate and infect naïve Vero cells. Subsequently, at 48 h.p.i, the Vero cells in the lower chamber were washed twice with ice-cold PBS and processed for another experiment. For the inhibition of exosome production, 20 μM of GW4869 inhibitor (Sigma-Aldrich, St. Louis, MO, USA), were also treated in this assay.

### 4.15. Statistical Analyses

GraphPad Prism version 8.0.2 for macOS (GraphPad Software, San Diego, CA, USA) was used for analyses of the statistically significant differences examined in the data sets. For a comparison of the two means of the entire data, the two-tail Student t-test of non-paired assumption was performed. All experiments were represented with Error bars indicating mean ± SD and *p* values. The *p* values of <0.05 were noted for the significance of statistical tests.

## 5. Conclusions

This is the first study to describe the characterization of secreted extracellular vesicles, including exosomes from CHIKV-infected epithelial cells, and their role in CHIKV transmission. The secreted vesicles from CHIKV-infected cells encapsulated CHIKV genomic RNA and any immunogenic parts of the virus. These EVs then played a role in producing infectious virions in other epithelial cells. These results might suggest the ability of EVs in the dissemination of CHIKV throughout host organs and further involvement in immune evasion inducing persistent arthritis (Figure 6).

## Figures and Tables

**Figure 1 ijms-23-12117-f001:**
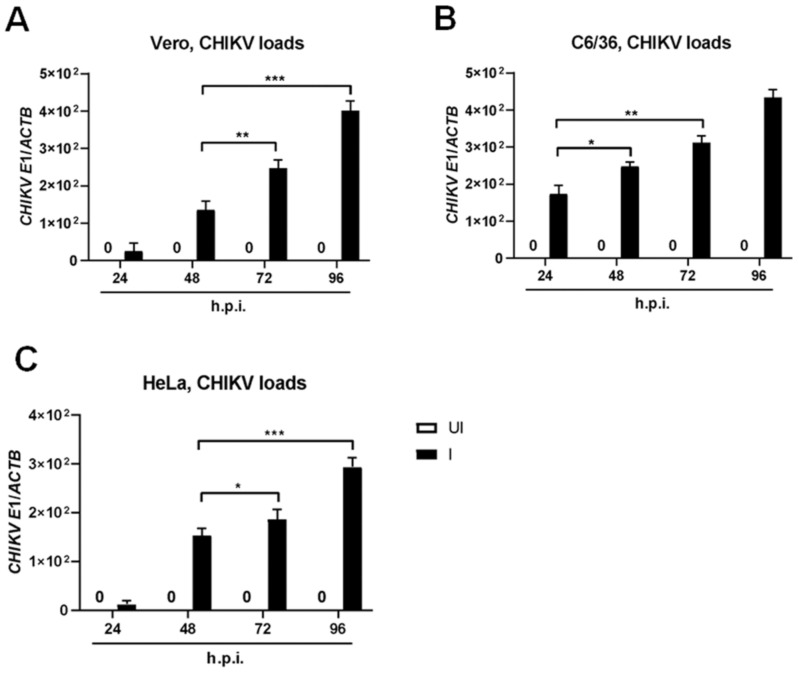
Kinetics infection of CHIKV ROSS C6D4 in various epithelial cell lines. qRT−PCR analysis showing transmembrane glycoproteins E1 mRNA loads in CHIKV-infected (MOI = 5) Vero cells (**A**), C6/36 cells (**B**), or in HeLa cells (**C**) at indicated time points. Each parental-uninfected cell type was used as a control. UI, CHIKV-uninfected cells; I, CHIKV-infected cells; *****, *p* < 0.05; ******, *p* < 0.01; *******, *p* < 0.0001.

**Figure 2 ijms-23-12117-f002:**
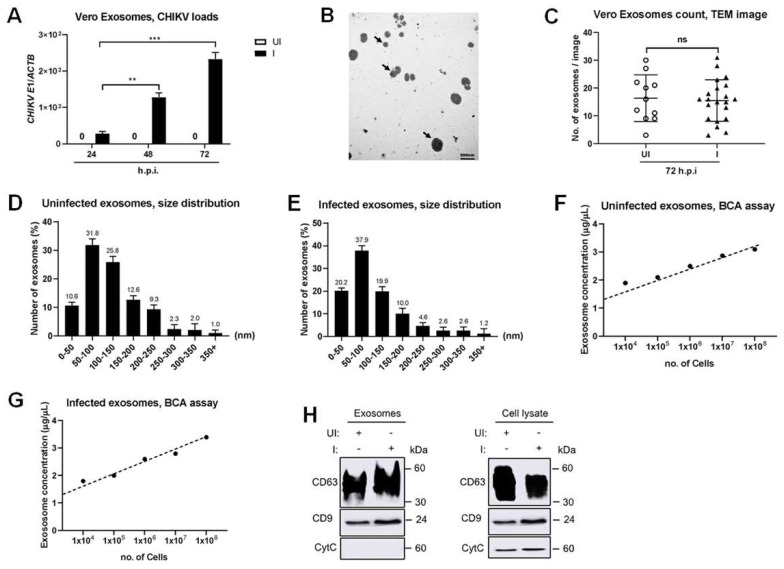
Physical characterizations and quantification of Vero−derived EVs with CHIKV infection. (**A**) Analysis of CHIKV loads in isolated exosomes derived from CHIKV−infected Vero cells at 5 MOI by qRT−PCR assay. (**B**) TEM images represented the exosome morphology that was isolated from CHIKV−uninfected (UI) or 5 MOI CHIKV−infected (I) Vero cells at 72 h.p.i.; Black arrow indicated isolated exosomes; Scale bar, 500 nm (magnification, 30,000×). (**C**) Counting of small EV numbers that were isolated from uninfected and 5 MOI CHIKV-infected groups at 72 h.p.i. The number of small EVs was represented on Y−axis. TEM images were gathered from at least three independent exosome isolation manners. The n indicates the total image number used in this analysis. (**D**,**E**) The distribution of the heterogeneous EVs size isolated from CHIKV−uninfected (**D**) or CHIKV−infected (**E**) Vero cells was demonstrated by the DLS technique. The number in the percentage and size range of each EV subgroup was indicated on Y−axis and X−axis, respectively. Based on the total number of small EVs in each size range, the percentages were calculated. (**F**,**G**) Total protein levels of Vero−derived small EVs from CHIKV−uninfected (**F**) or 5 MOI CHIKV−infected groups at 72 h.p.i. (**G**) were quantified by the BCA method. The total protein concentration of each small EV (Y−axis; measured as μg/μL) was plotted against the increase in Vero’s cell number (X−axis). In both groups, higher increments in total protein concentrations of small EVs progressively increased with the number of Vero cells. (**H**) CD63 and CD9, which are tetraspanin proteins and well recognized as exosome biogenesis markers, were detected on the exosomes isolated from the CHIKV−uninfected or 5 MOI CHIKV−infected Vero cells with the absence of the cytochrome−C (CytC), the negative exosome marker, in isolated exosomes by immunoblotting technique; ns, not significant; **, *p* < 0.01; ***, *p* < 0.0001.

**Figure 3 ijms-23-12117-f003:**
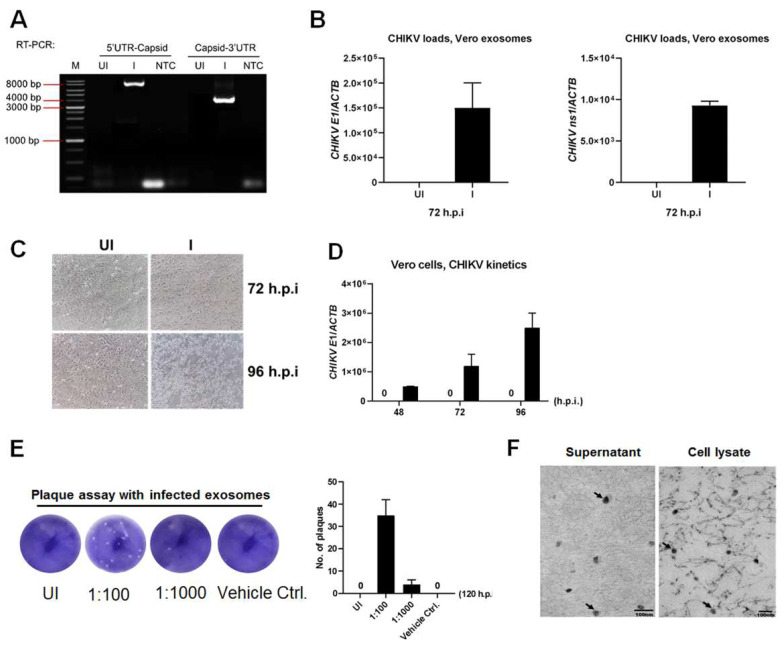
Exosomes derived from CHIKV−infected Vero cells contained viral genomic elements and are infectious. (**A**) An illustration of an agarose gel represents the overlapping amplicon product of all fragments from the entire CHIKV genome in exosomes derived from 5 MOI CHIKV−infected Vero at 72 h.p.i. The cDNA template was derived from the total exosomal RNA. All fragments display the amplification of the full−length CHIKV genome in each part. UI and I indicate CHIKV−uninfected and CHIKV−infected groups, respectively. N indicates no template control. M indicates DNA marker. (**B**) qRT−PCR analysis exhibiting CHIKV E1 mRNA loads in exosomes derived from MOI 5 infected Vero cells at 72 h.p.i. (**C**) An equal dose of exosomes from CHIKV−uninfected or CHIKV-infected Vero cells were applied to naïve Vero cells. Cytopathic effects (CPEs) were observed and daily noted via a bright-field inverted microscope. (**D**) qRT−PCR analysis revealing CHIKV loads in CHIKV−infected Vero cells at indicated time points. (**E**) Quantitative measurement of the plaque number on Vero cells treated with exosomes from CHIKV−uninfected or CHIKV−infected Vero cells. The column graph showed the CHIKV titer in the PFU/mL unit that was counted by PFU assay. (**F**) CHIKV was observed under a transmission electron microscope (TEM) from the cell culture medium and cell lysates of exosome infection naïve Vero cells. The black arrow indicated the CHIKV virions that were visual with a polymorphic giant form and many cores lining inside under TEM from both cell culture medium (**left**) and cell lysates (**right**) collected at 72 h.p.i. The virion size varied from 35 nm to 50 nm. The scale in both pictures was at 100 nm with 100,000× magnification.

**Figure 4 ijms-23-12117-f004:**
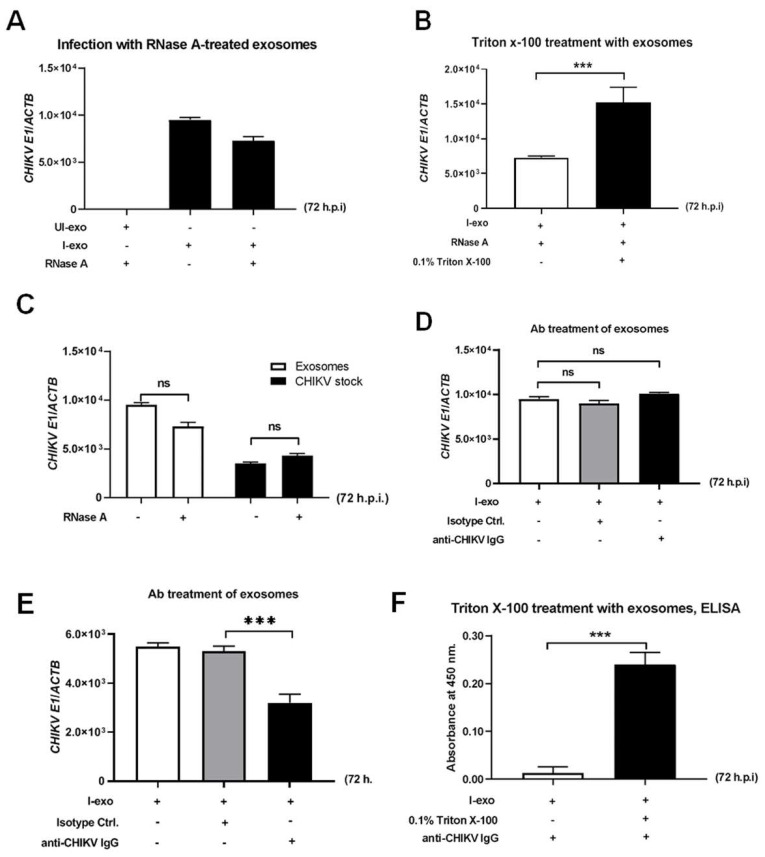
CHIKV RNA and proteins are securely contained inside small EVs. (**A**) qRT−PCR analysis representing CHIKV loads in naïve Vero cells that were incubated with exosomes derived from 5 MOI CHIKV accompanied by the RNase A treatment. Uninfected treated cells were used as control. (**B**) Relative qRT−PCR measurements from the exosomes fraction derived from 5 MOI CHIKV−infected Vero cells at 72 h.p.i. accompanied by the RNase A treatment with 0.1% Triton X−100 was demonstrated. The untreated group served as a control. (**C**) qRT−PCR analysis exhibiting CHIKV loads in naïve Vero cells infected via either infectious exosomes derived from 5 MOI CHIKV−infected Vero cells at 72 h.p.i. or 5 MOI of CHIKV laboratory viral stocks accompanied by the RNase A treatment. The untreated group served as a control. (**D**,**E**) qRT−PCR analysis revealed viral loads in exosomes derived from 5 MOI CHIKV−infected Vero cells at 72 h.p.i. (**D**) or 5 MOI of CHIKV laboratory viral stocks (**E**) that were treated with anti−CHIKV IgG from patient plasma or the matched isotype−control. Untreated exosome samples were used as the based-line control; ns indicates not significant. (**F**) ELISA assay representing the increased detection of CHIKV−derived proteins from exosomes isolated from 5 MOI CHIKV-infected Vero cells at 72 h.p.i. when treated with 0.1% Triton X−100 in comparison to untreated control; *******, *p* < 0.0001.

**Figure 5 ijms-23-12117-f005:**
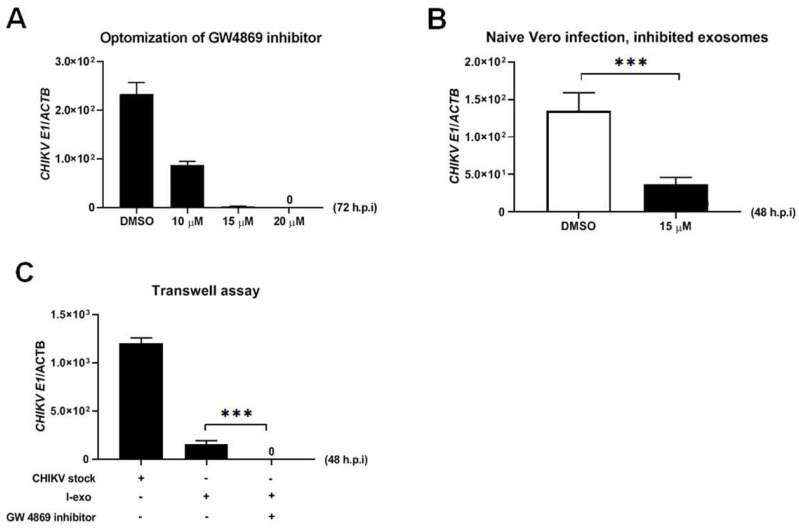
Role of the infectious EVs in CHIKV Cell−to−Cell transmission. (**A**) qRT−PCR analysis representing CHIKV loads in exosomes derived from DMSO− or GW4869−treated 5 MOI CHIKV−infected Vero cells at 72 h.p.i. (**B**) qRT−PCR analysis representing CHIKV loads in naïve Vero cells at 48 h.p.i., when treatment with exosomes derived from DMSO− or GW4869−treated 5 MOI CHIKV−infected Vero cells at 72 h.p.i. (**C**) qRT−PCR analysis representing CHIKV loads at 48 h.p.i. in naïve Vero cells in a transwell assay. We performed with Vero cells in the upper chamber and naïve Vero cells in the lower chamber treated with exosomes derived from 5 MOI CHIKV-infected Vero cells at 72 h.p.i. for 4 h in the presence or absence of 20 μM GW4869 inhibitor. Vero cells infected with CHIKV laboratory stocks with known titers were used as the based−line controls; ***, *p* < 0.0001.

**Figure 6 ijms-23-12117-f006:**
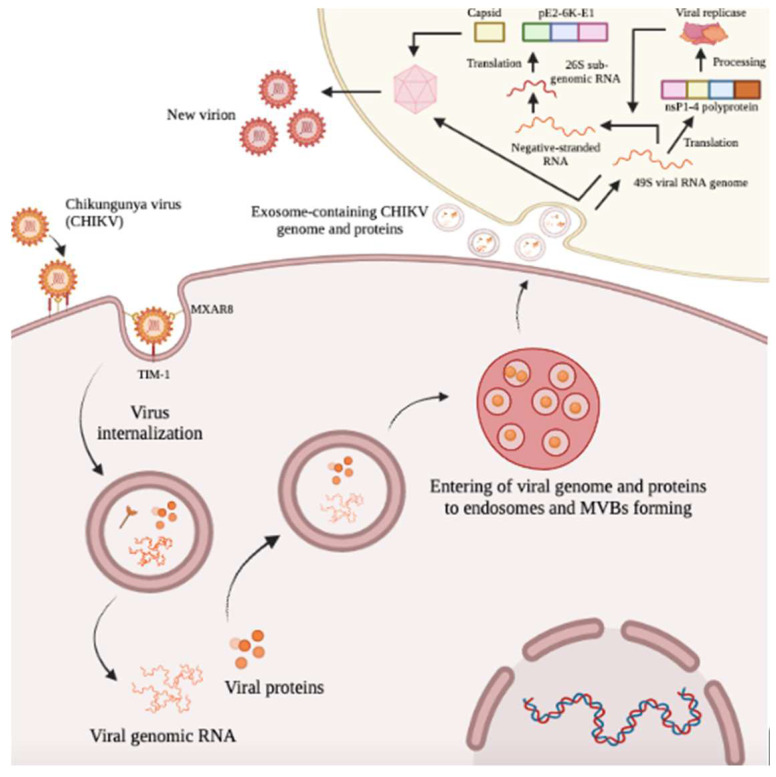
Schematic model of the involvement of exosomes during CHIKV infection. Visual representation of exosomes from CHIKV-infected epithelial cells participate in the Cell−to−Cell transmission of infectious viral elements. Figure created with the BioRender software (https://biorender.com/ accessed on 4 September 2022).

## Data Availability

Not applicable.

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
