# Peer review of "Characterization and Involvement of Exosomes Originating from Chikungunya Virus-Infected Epithelial Cells in the Transmission of Infectious Viral Elements"

_ijms, 2022, doi:10.3390/ijms232012117_

Round 1

Reviewer 1 Report

The manuscript by Le et al. reports a role of extracellular vesicles in CHIKV infection. The authors observed that CHIKV RNA and proteins were packaged into extracellular vesicles (EV) including exosomes with a size range from 50 to 250 nm by transmission electron microscopy. These EVs contained viral genome and proteins, which were able to replicate in naïve cells.  These EVs facilitated CHIKV transmission between cells and evaded CHIKV neutralizing antibodies.  Overall, the data are novel and interesting.  The manuscript is well written. 

The conclusions can be better substantiated and the impact of this manuscript can be improved by a few additional experiments.

1. Fig.4 F shows that the EVs contain viral proteins.  It would be important to characterize what specific viral proteins they are, structural or non-structural proteins?  They can do this by immunoblotting, include CHIKV stock and CHIKV-infected Vero cells as controls.

 2.  Fig. 5. Please clarify if GW4869 reduces infectious CHIKV virions in the cell culture supernatant.  

3.  The impact of this manuscript can be greatly improved if the in vitro results are validated in an animal model. The authors can simply compare the infection outcomes of CHIKV-EVs + isotype IgG and CHIKV-EVs + CHIKV-neutralizing IgG,  CHIKV stock + isotype IgG and CHIKV stock + CHIKV-neutralizing IgG. 

Author Response

Response to Reviewers #1

Manuscript ID: ijms-1934595

Title: Characterization and Involvement of Exosomes Originating from Chikungunya Virus-infected Epithelial Cells in the Transmission of Infectious Viral Elements

Point 1. Fig.4 F shows that the EVs contain viral proteins. It would be important to characterize what specific viral proteins they are, structural or non-structural proteins? They can do this by immunoblotting, include CHIKV stock and CHIKV-infected Vero cells as controls.

Response/ Revision 1: For the discussion about EVs containning viral proteins, many previous studies have shown the ability of neutralizing and immunogenicity of both non-structural protein (nsP2) [1, 2] and structural protein E1 and E2 [3-6]. We had the detailed description of the proteins that might be involved in the EVs from the main text lines 259-299. We also found the overlapped oligonucleotide amplicons that consisted of the entire CHIKV genome sequence in Vero cell-derived exosomes by a reverse transcription PCR (RT-PCR) method (figure 3A) as well as the CHIKV E1 mRNA (figure 3B). Additionally, the data of the neutralizing activity against small EVs fraction from CHIKV-infected Vero cells from patients’ plasma samples that presented a high titer of CHIKV positive human immunoglobulin G (IgG) and IgG negative control samples (figure 4D and E), also supported as a positive effect for the small group, suggested that some CHIKV proteins that are necessary for infection such as nsP2 and envelope proteins (E1, E2) securely contained inside the CHIKV-derived exosomes. From the above data, our study might suggest at least the presence of any part of the envelope protein (E1) internalized inside the EVs. Even though EVs could be affected by the human immune response, the internalized envelope protein might enhance the binding and fusion of EVs to adjacent cells, thereby facilitating the spread of infection.

Point 2. Fig. 5. Please clarify if GW4869 reduces infectious CHIKV virions in the cell culture supernatant.

Response/ Revision: GW4869 as we mentioned in our experiment, a dihydrochloride hydrate substance, is a selective cell-permeable neutral sphingomyelinase inhibitor, affecting the biogenesis and releasing of exosomes from the cells. In our experiment on the role of infectious EVs in CHIKV cell-to-cell transmission (figure 5), we got significantly decreased CHIKV loads in the small EV fraction treated with 15 μM of GW4869. This information indirectly showed the decrease of viral replication that not only reduces the transmission of the virus from cell to cell but might indirectly reduce the virus in the cell culture supernatant.

Point 3. The impact of this manuscript can be greatly improved if the in vitro results are validated in an animal model. The authors can simply compare the infection outcomes of CHIKV-EVs + isotype IgG and CHIKV-EVs + CHIKV-neutralizing IgG, CHIKV stock + isotype IgG and CHIKV stock + CHIKV-neutralizing IgG.

Response/ Revision: The validation of these experiments on animal models will be great and prospective experiments for clearly understanding and confirmation all our hypotheses on the mechanism of CHIKV infection and transmission via exosome. We hope to perform this experiment in the future for a completed study.

References

  1. Meshram, C. D., T. Lukash, A. T. Phillips, I. Akhrymuk, E. I. Frolova, and I. Frolov. "Lack of Nsp2-Specific Nuclear Functions Attenuates Chikungunya Virus Replication Both in Vitro and in Vivo." Virology 2019, 534: 14-24. doi:10.1016/j.virol.2019.05.016
  2. Zhang, Y. N., C. L. Deng, J. Q. Li, N. Li, Q. Y. Zhang, H. Q. Ye, Z. M. Yuan, and B. Zhang. "Infectious Chikungunya Virus (Chikv) with a Complete Capsid Deletion: A New Approach for a Chikv Vaccine." J Virol 2019, 93, no. 15. doi:10.1128/jvi.00504-19
  3. Bagno, F. F., L. C. Godói, M. M. Figueiredo, S. A. R. Sérgio, T. F. S. Moraes, N. C. Salazar, Y. C. Kim, A. Reyes-Sandoval, and F. G. da Fonseca. "Chikungunya E2 Protein Produced in E. Coli and Hek293-T Cells-Comparison of Their Performances in Elisa." Viruses 2020, 12, no. 9. doi:10.3390/v12090939
  4. Masrinoul, P., O. Puiprom, A. Tanaka, M. Kuwahara, P. Chaichana, K. Ikuta, P. Ramasoota, and T. Okabayashi. "Monoclonal Antibody Targeting Chikungunya Virus Envelope 1 Protein Inhibits Virus Release." Virology 2014, 464-465: 111-17. doi:10.1016/j.virol.2014.05.038
  5. Schnierle, B. S. "Cellular Attachment and Entry Factors for Chikungunya Virus." Viruses 2019, 11, no. 11. doi:10.3390/v11111078
  6. Tumkosit, U., U. Siripanyaphinyo, N. Takeda, M. Tsuji, Y. Maeda, K. Ruchusatsawat, T. Shioda, H. Mizushima, P. Chetanachan, P. Wongjaroen, Y. Matsuura, M. Tatsumi, and A. Tanaka. "Anti-Chikungunya Virus Monoclonal Antibody That Inhibits Viral Fusion and Release." J Virol 2020, 94, no. 19. doi:10.1128/jvi.00252-20

Reviewer 2 Report

This is an interesting paper that is highly original. There is, however, room for improvement:

1) There is a need for a more thorough description of exosomes especially with regard to its function. More can be found at:

https://cellandbioscience.biomedcentral.com/articles/10.1186/s13578-019-0282-2

The authors could pay attention to the passage in the above paper:

"Exosomes are nano-sized biovesicles released into surrounding body fluids upon fusion of multivesicular bodies and the plasma membrane. They were shown to carry cell-specific cargos of proteins, lipids, and genetic materials, and can be selectively taken up by neighboring or distant cells far from their release, reprogramming the recipient cells upon their bioactive compounds."

2 Notice the words "body fluids" above, The mosquitoes are likely to carry the virus  while exposing it to its saliva as in dengue virus and many flaviviruses. An interesting article is that of rabie, which resides near the salivary gland of animals such as dogs.  And the rabies virus use exosomes for its transmission between cells:

https://www.ncbi.nlm.nih.gov/pmc/articles/PMC6420551/

3) Looking the data from Figure 1, obviously, the virus prefers certain cell types. More about the cell types tha CHIK prefers can be found at:

https://pubmed.ncbi.nlm.nih.gov/22686853/

5)  I am very curious about the results seen in Figure 1. It  is basically saying that CHIKV prefers VEROS Cells and C6/36 to Hela cells. The question is why/ Could it be that VEROS cells are more associated with bodily fluids as mentioned in (1) and (2), compared to Hela Cells since Veros was from the kidney cells of green monkeys??  Do you have any data pertaining to the amount of exosomes in each fo the cell types?  If you do, it can make the paper even stronger.

3) "Total exosomes were purified"

is abit awkwar. Better to say " Much of the exosomess..." or "All of the exosomes.."

4) "estimated the total protein"

Did they mean "total amount  of proteins"

5)Similaryly, "

had the total protein"

Did they mean "total amount of proteins"

Author Response

Response to Reviewers #2

Manuscript ID: ijms-1934595

Title: Characterization and Involvement of Exosomes Originating from Chikungunya Virus-infected Epithelial Cells in the Transmission of Infectious Viral Elements

Point 1. There is a need for a more thorough description of exosomes especially with regard to its function. More can be found at:

https://cellandbioscience.biomedcentral.com/articles/10.1186/s13578-019-0282-2

The authors could pay attention to the passage in the above paper:

"Exosomes are nano-sized biovesicles released into surrounding body fluids upon fusion of multivesicular bodies and the plasma membrane. They were shown to carry cell-specific cargos of proteins, lipids, and genetic materials, and can be selectively taken up by neighboring or distant cells far from their release, reprogramming the recipient cells upon their bioactive compounds."

Response/ Revision: We will add in the main manuscript some information from the above interesting paper from Zang et al. (Ref: “Exosomes: biogenesis, biologic function and clinical potential - Yuan Zhang, Yunfeng Liu, Haiying Liu* and Wai Ho Tang*). The added information had been highlighted and well noted in the main manuscript.

Point 2. Notice the words "body fluids" above, the mosquitoes are likely to carry the virus while exposing it to its saliva as in dengue virus and many flaviviruses. An interesting article is that of rabie, which resides near the salivary gland of animals such as dogs. And the rabies virus use exosomes for its transmission between cells:

https://www.ncbi.nlm.nih.gov/pmc/articles/PMC6420551/

Response/ Revision: I will explain the reason why we choose the Vero cell in our study in the below response.

Point 3. Looking the data from Figure 1, obviously, the virus prefers certain cell types. More about the cell types tha CHIK prefers can be found at:

https://pubmed.ncbi.nlm.nih.gov/22686853/

Response/ Revision: The suggested paper from Tang about “The cell biology of Chikungunya virus infection” is a useful reference for my manuscript. I would like to select some information to add to our manuscript. The added information had been highlighted and well noted in the main manuscript.  

Point 4. I am very curious about the results seen in Figure 1. It is basically saying that CHIKV prefers VEROS Cells and C6/36 to Hela cells. The question is why/ Could it be that VEROS cells are more associated with bodily fluids as mentioned in (1) and (2), compared to Hela Cells since Veros was from the kidney cells of green monkeys?? Do you have any data pertaining to the amount of exosomes in each fo the cell types? If you do, it can make the paper even stronger.

Response/ Revision: We have a detailed explanation for choosing the Vero cell for this study:

  1. We isolated EVs from CHIKV- infected Vero cell since the Vero cell line met the requirements for both susceptible and epithelial cells, one of the favored tropism cells of CHIKV in the human body and the target of many kinds of research on the pathogenesis of CHIKV. Some previous studies also demonstrate that Vero cells were sensitive to CHIKV infection [1] or the ability ECSA lineages (especially ROSS strain - the control CHIKV used in this study) showed high susceptibility to both infection and apoptosis with rates of over 80% by 5 d.p.i. on Vero cells meanwhile the C6/36 cell line (Aedes albopictus cell line) does not show any susceptibility with this strain of CHIKV [2]. In addition, a study by Li et al. (2013) on the CHIKV Ross strain also concluded that CHIKV infection induced strong CPE and apoptosis in the Vero cells but light CPE in the C6/36 cells [3].
  2. Taking from our data, the susceptibility profiles of the established epithelial cell lines to CHIKV infection showed that CHIKV at the multiplicity of infection (MOI) 5 facilely infected the Vero cell line with significantly increased transmembrane glycoproteins E1 mRNA loads at 72 hr. and 96 hr. post-infection. In addition, the amount of Vero cell-derived exosomes carried CHIKV transmembrane glycoproteins E1 mRNA loads were highest at 72 h.p.i (figure 2A), which is the reason for selecting for the isolation of exosomes from CHIKV-infected epithelial cell.
  3. We don’t have data on the number of exosomes for each type of cell line since we already chose the Vero cell for our study in the first step.

Point 5. "Total exosomes were purified" is abit awkwar. Better to say " Much of the exosomess..." or "All of the exosomes."

Response/ Revision: Change to the new term “much of the exosomes” (line 125)

Point 6. "estimated the total protein"

Did they mean "total amount of proteins"

Response/ Revision: Change to the suggested words “total amount of proteins” (line 147)

Point 7. Similaryly, "had the total protein"

Did they mean "total amount of proteins"

Response/ Revision: Change to the suggested words “total amount of proteins” (line 148)

References:

  1. Sourisseau, M., C. Schilte, N. Casartelli, C. Trouillet, F. Guivel-Benhassine, D. Rudnicka, N. Sol-Foulon, K. Le Roux, M. C. Prevost, H. Fsihi, M. P. Frenkiel, F. Blanchet, P. V. Afonso, P. E. Ceccaldi, S. Ozden, A. Gessain, I. Schuffenecker, B. Verhasselt, A. Zamborlini, A. Saïb, F. A. Rey, F. Arenzana-Seisdedos, P. Desprès, A. Michault, M. L. Albert, and O. Schwartz. "Characterization of Reemerging Chikungunya Virus." PLoS Pathog 2007, 3, no. 6: e89. doi:10.1371/journal.ppat.0030089
  2. Wikan, N., P. Sakoonwatanyoo, S. Ubol, S. Yoksan, and D. R. Smith. "Chikungunya Virus Infection of Cell Lines: Analysis of the East, Central and South African Lineage." PLoS One 2012, 7, no. 1: e31102. doi:10.1371/journal.pone.0031102

3.         Li, Y. G., U. Siripanyaphinyo, U. Tumkosit, N. Noranate, A. nuegoonpipat A, R. Tao, T. Kurosu, K. Ikuta, N. Takeda, and S. Anantapreecha. "Chikungunya Virus Induces a More Moderate Cytopathic Effect in Mosquito Cells Than in Mammalian Cells." Intervirology 2013, 56, no. 1: 6-12. doi:10.1159/000339985  

Round 2

Reviewer 1 Report

No further questions. 

Reviewer 2 Report

Improvements seen in this version.